# Which turtle should I study? Uneven distribution of research effort across Testudines species

Simon Ducatez[1]*, Jayna Lynn DeVore[2]

**1** UMR 241 SECOPOL (IRD, IFREMER, ILM, UPF), Punaauia, Tahiti, French Polynesia, **2** UMR 241 SECOPOL (UPF, IRD, IFREMER, ILM), Punaauia, Tahiti, French Polynesia

* simon.ducatez@gmail.com

## Abstract

Understanding why some species are more often studied than others can provide valuable information on potential biases that research effort discrepancies can cause and help to direct future research to mitigate these biases. Turtles form an order of reptiles in which the majority of species (67%) are threatened with extinction, but they show high variation in the amount of research devoted to each species. We aimed to quantify this variation across all recognized extant species, and to decipher how the distribution of this research effort varied according to species taxonomy, phylogeny, distribution, ecology, life history and extinction risk. Using the number of references listed on the Web of Science for each species as a proxy for species research effort, we first show that this index is both robust to variation in search methods within WoS, and consistent with an index extracted from another online library (Scopus). The number of articles per species varied substantially, with 3441 articles for the green sea turtle alone, but only 475 total articles for all 50% least studied species taken together. This heterogeneity was consistent across time since the 1970s. Phylogenetic non-independence explained 66% of the variance in research effort (vs 58% for taxonomy). In addition, marine species, a group that only includes seven species, were particularly highly studied: five of the eight most studied Testudines are marine turtles. In contrast, freshwater and terrestrial species shared similar research effort values. Species occurring in Europe and North America were also more studied than species from, e.g., Central America or Africa, as were species that occur in the most scientifically active countries. In addition, species with a larger distribution range or introduced outside of their native range, as well as species with certain life history characteristics like larger clutches, smaller eggs or a longer longevity were more frequently investigated. In contrast, population trend and extinction risk did not predict research effort, although species that have not been assessed by the IUCN were characterized by a low research effort. This study underlines the importance of accounting for the heterogeneity in research effort across turtle species in global

**Data availability statement:** All relevant data are within the manuscript and its Supporting Information files.

**Funding:** Agence nationale de la recherche, France InvEcoF #ANR-22-CE02-0007 Fondation Fyssen, France. The funders had no role in study design, data collection and analysis, decision to publish, or preparation of the manuscript.

**Competing interests:** The authors have declared that no competing interests exist.

analyses and helps to identify taxa, regions, and life history strategies that should be the focus of more research.

## Introduction

Particularly sensitive to the current extinction crisis [1,2], Testudines are one of the most threatened groups of vertebrates: 67% of the assessed extant species are threatened with extinction [3], as compared to 25% of mammals, 20% of squamates and 14% of birds. Although several threats are responsible for this pattern, wildlife trafficking (for pet trade, turtle meat or body parts) was identified as the most damaging threat driving many species towards extinction [2]. Often referred to as the "Asian turtle crisis" because of how important this threat is in Asia [4,5], it has recently been named a "Global turtle crisis" [6] given the universal nature of turtle trafficking. Testudines occupy terrestrial, freshwater and marine ecosystems, where they participate in a diversity of ecosystem functions, including herbivory, seed dispersal and detritivory [7–11]. Their functional role in some ecosystems is sufficiently important that tortoises have become a model group in ecosystem restoration via rewilding, including through taxon substitution [8,10,12–16]. Understanding the ability of Testudines to respond to current environmental changes is therefore important to better preserve not only the turtles and tortoises themselves but also the functioning of the ecosystems they are part of.

Although some particularly iconic species draw a lot of public interest and have attracted an important amount of research (e.g., marine species), many species remain poorly known. This knowledge gap is likely to impact our ability to globally understand the ecological roles of turtles and tortoises, the capacity of this unique taxonomic group to respond to current biodiversity and climate crises, and our ability to preserve this group. Thanks to the efforts of multiple researchers, and especially of the Turtle Taxonomy Working Group [17,18], the current knowledge of Testudines species has been assembled, and the risk of extinction of 70% of the 359 currently recognized extant turtle species has now been assessed [3]. However, the conservation status of the remaining 30% (110 species) remains unclear. This reflects the existence of important discrepancies in the amount of research that has been dedicated to the different species of Testudines.

Knowledge gaps and discrepancies in research effort can affect our understanding of the ecology, evolution, and conservation of Testudines for several reasons. First, they lead to the exclusion of the least studied species from reviews and comparative analyses, making the conclusions of these global studies less representative and more difficult to interpret. Second, they can affect trait estimates, e.g., because extreme values are more likely to be recorded in more investigated species. Third, they can alter the accuracy of species trait value estimates. Such values may be biased towards one or few individuals or populations in understudied species, but more representative of the entire species in more studied species. Finally, these knowledge gaps directly affect researchers and experts' ability to assess the conservation status of understudied species, excluding them *de facto* from conservation programs.

Such research gaps are common across taxonomic groups and have been identified across a diversity of taxa. Predictors of taxonomic variation in research effort have for example been investigated in birds [19], mammals [20,21], fishes [22,23], amphibians [24] and plants [25]. Recently, a global assessment of variation in research effort across reptile species was conducted [26], revealing important variation across reptile groups, and identifying biological and socioeconomic factors that determine how often a given species is studied. Such broad scale comparative analyses are particularly useful to identify factors determining discrepancies in research effort across species, and help mitigating these. Guedes et al. (2023) [26] also brought to light an important fact about Testudines: although this class emerges as one of the most studied on average among reptiles, it also shows extreme variation within the group, with families such as the Cheloniidae having attracted the most attention while others such as Chelydridae were much less studied [26]. These observations call for a more detailed investigation of variation in research effort across Testudines species, an aspect that was not investigated in this previous work.

The objective of this study is to provide such investigation. On the one side, we aimed to identify which species and families are particularly studied, and which ones are particularly neglected, and at deciphering whether this relates to the ecology, life history and distribution of these species. For example, are aquatic turtles more often studied than terrestrial tortoises? Are species with a fast life history (early reproduction and shorter lifespan) more studied than species with a slow life history (late age at first reproduction, high longevity)? How does research effort vary across biogeographic realms in Testudines? Is research effort targeting species that have a higher risk of extinction? By clearly identifying understudied species, we hope to motivate the funding of research programs aiming at filling these knowledge gaps. In parallel, we aimed to determine how research effort variation may affect our global understanding of the ecology, evolution and conservation of turtles and tortoises. Literature reviews are necessary to understand global patterns and mechanisms within entire taxonomic groups such as the Testudines. However, they are also limited by the available literature and data: if research is taxonomically biased, the conclusions drawn from these presumably global reviews are likely to also be taxon-specific, making it important to identify these biases. Differences in research effort may also have direct consequences for some trait estimates. For example, turtles and tortoises are famous for their long life expectancy, with records in some species exceeding human lifetimes (> 100 years in several giant tortoise species, [27]). Record longevities are however more likely to be detected in intensively studied species, as compared to less studied ones. Progress in our knowledge of Testudines phylogeny (there is now a tree based on genetic data from 80% of all turtle species: [28]), together with advances in statistical analyses now allow researchers to run large scale analyses considering most species in this order and to bring a more global understanding of the importance of this group, but also of the threats it is facing. Identifying knowledge gaps should help taking them into account when designing these global analyses and interpreting their results.

Here, we first provide an index of research effort for each extant Testudines species, available for use in comparative studies. We tested for its robustness by comparing it with other indices. Second, we investigated how spread out this research effort is, and tested whether research discrepancies improved or, on the contrary, became more extreme across time. Third, we investigated whether variation in research effort was related to taxonomy and phylogeny, or to 14 variables describing the distribution range, ecology, and life history of turtles. Finally, we tested whether research effort was predicted by extinction risk and population trend.

## Methods

### Research effort estimation

We used the taxonomy from the Turtle Taxonomy Working Group 2025 [18], except for the Galapagos tortoises (*Chelonoidis sp.*) for which we followed the taxonomy considered by the IUCN, which recognizes 12 species that are considered as 12 subspecies of *C. niger* by [18]. This allowed to include this important taxon in the analyses considering extinction risk (the IUCN provides 12 different assessments for this set of 12 taxa) and resulted in a total of 370 extant species.

We estimated the research effort devoted to each one of these 370 species by extracting the number of references listed in the Web of Science (WoS) database (extraction conducted in January 2024). For each species, we conducted a search on WoS for papers published between 1975 and 2023, using the current species scientific name (e.g., "*Gopherus morafkai*") as keyword, and considering *all fields* (not just the title or keywords). We then extracted the number of articles obtained in the search as the index of research effort. In parallel, we conducted the same search but considering all synonyms of each species as keywords using the synonyms listed in [18] (e.g., "*Gopherus morafkai*" OR "*Xerobates morafkai*"), to determine how name changes affected the research effort estimates. Progress in taxonomy can lead to multiple changes in species names, so that only including the current name may underestimate the research effort for some species [29]. We also ran these two searches again but focusing on titles only, to compare an index of research effort based on the number of papers that are most likely to be focused on the focal species (as its name is in the title) with a more inclusive index that also considers articles including the focal species name in the abstract or keywords. Even in an article focused on a given species, the name of the species may not appear in its title, but in the keywords or abstract. In contrast, a species name may appear in the abstract or keywords of an article even if the article is not focused on that species. Comparing the research effort considering titles only vs all fields should allow us to determine whether changing the search method affects the resulting indices. Note that these indices are proxies for research effort, and not absolute estimates. They, for example, ignore the work that was published before 1975 as well as any reference that is not referenced in the WoS, including references from the grey literature. However, by constructing these indices, our purpose was to obtain a relative metric of research effort to compare species with each other, rather than an exact number of articles published per species. To test whether using different databases could affect this relative research effort score, we also compared the index obtained using the WoS database with the index extracted by [26] using Scopus and available for 343 of the 370 species considered.

## Variation in research effort distribution across time

To assess variation in research effort across time and species, we extracted the number of papers referenced in the WoS database for each species and decade (i.e., from 1980 to 1989, from 1990 to 1999, from 2000 to 2009, and from 2010 to 2019). For this approach, we included all fields (not just the title) and considered all synonyms of a given species name as keywords to be more inclusive and to account for name changes across time.

## Trait collection

We extracted values for 14 variables likely to either affect research effort or be affected by research effort: biogeographic realm, country research rank, range size, insularity, habitat, successful introduction, diet, carapace length, maximum longevity, generation length, female age at maturity, clutch size, incubation time and egg length. We extracted information for each species on biogeographic realm (ten different regions and one category for species occupying two or more regions), range size (both the estimated indigenous (historical) range (AOO) from [18] and current range size available in [26]), habitat (marine, terrestrial, freshwater or mixed for species occurring in both terrestrial and freshwater habitats), insularity (island or continent) and diet (herbivorous, carnivorous or omnivorous) from [3,17,18,27]. Information on whether successful introductions of each species have occurred was also available in [18]. Country research rank was scored for each species based on the species distribution and Scimago country ranking in November 2025 (https://www.scimagojr.com/countryrank.php). The Scimago country rank is based on the number of articles published per country between 1996 and 2024. Species present in at least one of the top 10 countries were given a score of 3, a score of 2 if present in at least one of the top 10–20 countries, or a score of 1 if not present in any of the top 20 research producing countries. We considered the maximum straight-line carapace length (in cm) as body size metric, available for most species from [17,18]. In addition, we retrieved information on maximum longevity, generation length, female age at maturity, clutch size, incubation

time and egg length from https://www.demogr.mpg.de/longevityrecords/0403.htm, [27,30–32]. We extracted the risk of extinction and population trend of each assessed species from the IUCN's web site [3] in November 2025.

## Analyses

We ran all the analyses using R [33]. To compare the different indices of research effort, we used Pearson correlation tests and coefficients on each pair of log-transformed indices (considering log (research effort +1)). To characterize the level of aggregation of research effort, we used the Poulin index of discrepancy (D, [34]). It allowed us to measure the deviation of research effort distribution from a theoretically even distribution among species. D ranges from zero to one, indicating no aggregation (0) to highly aggregated (1) distribution. This index was originally developed to quantify parasite aggregation across individuals in a host population. When investigating changes in D across decades, we excluded the species that had not yet been described in 1980. Research can only be dedicated to a given species after its description, and we aimed at using the same set of species to make the comparison meaningful. However, for the global estimate of discrepancy, we included all species. We used the QPweb software to calculate D [35].

## Phylogeny and taxonomy

We built a Phylogenetic Linear Mixed Model (PLMM) using the function MCMCglmm from the R package *MCMCglmm* [36] to estimate the proportion of variance in research effort (log-transformed) explained by phylogeny. We included phylogeny as random effect in a model with research effort (log-transformed) as response variable, using the phylogeny from Thomson *et al.* (2021), which used nuclear DNA to infer the phylogenetic relationships of 80% of the recognized extant Testudines species. This analysis was therefore conducted on a subset of 275 species included in this phylogeny. We used a poorly informative inverse Wishart prior ($V = 1$, $\nu = 0.002$) for the variances, following [36], and ran the model for 550001 iterations with a burn-in interval of 50000 to ensure satisfactory convergence, sampling one every 500 iterations, and checking that autocorrelation levels among samples was lower than 0.1. We assessed chain convergence by visually analyzing trace plots and running Gelman-Rubin convergence diagnostic on five chains (*gelman.diag* function from the package *coda*, [37]). We calculated the intra-class coefficient (ICC) using the variance estimates of the model (with $ICC = Vp/ (Vp + Vr)$, where $Vp$ = phylogenetic variance and $Vr$ = residual variance).

To determine the proportion of variance in research effort explained by taxonomy, we built linear mixed models (LMM) with research effort (log + 1 transformed) as response variable, and either family or genus nested into family as random effects, using the *lme* function in the R package nlme [38]. We also built a null model (using the gls function) with no random effect and compared the AIC of these three models. We calculated the intra-class coefficient (ICC) of each taxonomic level using the variance estimates provided by the full model for both the family and genus effects.

## Predictors of research effort

We investigated the association between research effort and 14 species characteristics, including biogeographic range, country research rank, range size (considering either the estimated indigenous (historical) range or current range size), insularity (binary variable separating island from continent species), habitat (factor with four levels: marine, freshwater, terrestrial or both freshwater and terrestrial), introduction (binary variable segregating successfully introduced species from other species), diet (herbivorous, carnivorous, omnivorous), maximum straight-line carapace length (SCL), maximum longevity, generation length, female age at maturity, clutch size, egg length and incubation time. We also ran an extra analysis focusing on either freshwater or terrestrial species to test whether the research effort varied between freshwater species occurring in rivers, still waterbodies or both rivers and still waterbodies, and between terrestrial species occurring in forests, open habitats, or both. We first tested for direct associations between research effort and each trait independently using Spearman rank correlations for continuous or ordinal variables (as some variables were not normally

distributed), Wilcoxon tests for binary variables, or building linear models with research effort (log-transformed) as response variable and the factor of interest as explanatory variable for categorical variables (biogeographic range, habitat and diet). This set of analyses allowed to maximize the sample size, as all the species with available information for the trait of interest could be included.

We then tested whether these associations were consistent when controlling for phylogeny (or whether phylogenetic non-independence alone could explain these associations), building one LMM per predictor with research effort (log-transformed) as response variable, the predictor of interest as explanatory variable and phylogeny as random effect. This analysis restricted the dataset to the 275 species included in the phylogeny. For categorical variables, we compared the deviance information criterion (DIC) of the model including vs excluding the categorical variable of interest to assess whether it was associated with research effort. The DIC is a variation of the AIC used in Bayesian model selection using MCMC simulations, and lower DIC indicate models with a better fit. Finally, we included all the predictors as fixed effects into one LMM with phylogeny as random effect to determine which predictors of research effort were the most import-ant. We used a stepwise selection, removing non-significant predictors one by one until all predictors were significant ($p$MCMC < 0.05). We also used DIC comparisons of models including vs excluding each categorical variable to decide whether to include them or not in the final model (as MCMCglmm does not provide unique $p$MCMC values for categorical variables). Once a final model was selected, we added each removed parameter one by one to the model to validate it, determining whether previously removed variables were still non-significant (or resulted in models with a higher DIC, using a threshold of $\Delta$DIC = 5). For this analysis, we scaled all continuous predictors to a mean of 0 and a variance of 1 to obtain comparable estimates. This final global model could only include the species with trait values available for all the traits considered, and therefore was conducted on a significantly decreased sample size. Female age at maturity and genera-tion length strongly decreased the number of species included in this analysis, and these factors were never significantly associated with research effort, so we excluded them from this last analysis to maximize the sample size. The correlation between predictors was always <0.4 except between maximum SCL and clutch size (Spearman $\rho$ = 0.568, p < 0.001). To avoid collinearity issues, we excluded clutch size from this analysis. With regards to range size, we considered the esti-mated indigenous range size (AOO), available for all species in our dataset, since current range size was only available for 313 of the 370 species (the two metrics were highly correlated: Spearman $\rho$ = 0.931). The dataset for this multivariate analysis considered the 115 species for which all 11 traits were available. We reran this model selection after excluding the sea turtles to test for result consistency.

### Extinction risk

We tested whether the research effort for IUCN-assessed species differed from the non-assessed species. Then, we tested whether extinction risk was associated with research effort, ranking the five IUCN extinction risk categories from 1 (critically endangered) to 5 (least concern) and excluding data deficient, extinct and non-assessed species [3]. We also tested whether threatened (CR, EN or VU) and non-threatened (NT or LC) species had different research efforts, and whether population trend (with 0 = decreasing, 1 = stable or 2 = increasing, available from [3]) predicted research effort. To that aim, we built linear models with research effort (log-transformed) as response variable and either IUCN assessment (assessed or not), extinction risk, threatened (yes/no) or population trend as fixed effects. Considering extinction risk and population trend as continuous or as categorical variables resulted in the same conclusions so we only present the results considering them as continuous variables.

### Results

Estimating research effort using species names in titles only or in titles, abstracts and keywords provided highly correlated coefficients (Pearson correlation coefficient between the log-transformed indices = 0.966; p < 0.001). Similarly, considering synonyms or focusing on current names resulted in highly correlated coefficients (Pearson = 0.973; p < 0.001). The research

effort extracted from Scopus by [26] was also highly correlated with the indices obtained with references listed in the WoS (e.g., when considering all fields and synonyms, Pearson = 0.953; p < 0.001; n = 342 species). In the following analyses, we used the index considering species synonyms and all fields (not just the title) in the search for papers on WoS.

The search resulted in a total of 24442 articles on the Web of Science. The mean number of articles per species was 66 (SE = 16), and the median was six articles (range = 0–3441), suggesting that the distribution was highly right-skewed. The Poulin index of discrepancy confirmed that the distribution of research effort was highly aggregated (D = 0.873, CI = [0.835; 0.912]). Although there was a small decrease in D across time (Poulin D; 1980s: 0.904, CI = [0.877 to 0.932]; 1990s: 0.905, CI = [0.869; 0.929]; 2000s: 0.867, CI = [0.830; 0.900]; 2010s: 0.860, CI = [0.809 to 0.899]), the change in discrepancy across decades was minimal (from 0.904 to 0.860).

The green sea-turtle (*Chelonia mydas*, with 3441 articles) and the loggerhead sea-turtle (*Caretta caretta*, with 3334 articles) were the two species with the highest index of research effort, followed by the pond slider (*Trachemys scripta*, with 2157 articles), painted turtle (*Chrysemys picta*, with 1445 articles) and common snapping turtle (*Chelydra serpentina*, with 1082 articles). The next three species were all marine turtles, so that out of the seven extant species of marine turtles, five are among the eight most studied species of Testudines. Sea turtles represent 1.9% of all Testudines species but gathered 39% of all publications in this group, with the least studied sea turtle (*Natator depressus*) obtaining a research effort score of 105 articles, 17 times higher than the median score for all Testudines. The most studied terrestrial species was the desert tortoise (*Gopherus agassizii*, ranking 11th with 454 articles). Thirty-two species had a score of 0 (see Suppl. Mat. 4), including species such as the enigmatic leaf turtle *Cyclemys enigmatica*, while the 50% least studied species gathered only 2% of the global research effort (475 articles for 185 species). Across decades, the number of articles published on turtles increased sharply: from 1067 in the 1980s to 9766 in the 2010s, i.e., 9.2 times more in the 2010s than in the 1980s (Fig 1). This increase was particularly important in marine turtles, which attracted 18.1 times more research in the 2010s as compared to the 1980s (Fig 1).

## Taxonomic and phylogenetic variation

The model including genus nested in family (AIC = 1329) had a better fit than the one including family only (AIC = 1346) or the null model (AIC = 1402). Together, family and genus explained 58% of the variance in research effort (ICC = 0.49

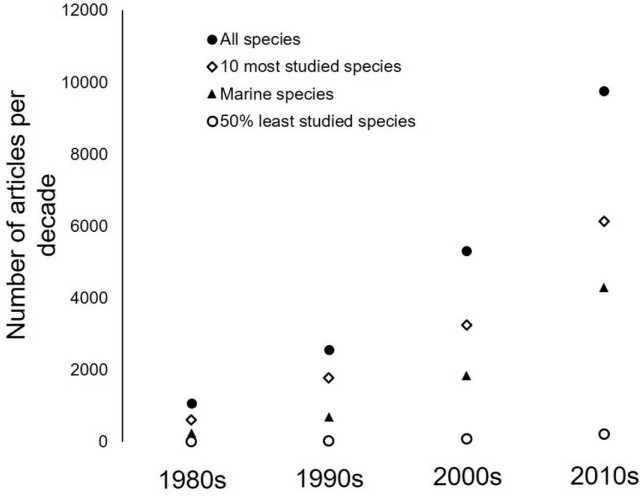

**Fig 1. Evolution of the number of articles on turtle species across time for all turtle species, for the 10 most studied species, for marine species only, and for the 50% least studied species.**

for family, ICC = 0.09 for genus). Phylogeny accounted for 66% of the variance when including it as a random effect in a PGLMM considering the 275 species of Testudines included in the phylogeny (extracting the ICC for family and genus with this same subset of 275 species resulted in an ICC of 0.48 for family and 0.09 for genus – thus 57% in total).

More diverse families also attracted more research (the number of papers per family increased significantly with the number of species per family, Fig 2). However, species diversity within a given family was not significantly correlated with the mean research effort per species (Fig 2). Some families were investigated particularly frequently, especially the two sea turtle families, while others were particularly neglected, such as the Pelomedusidae, a family of African freshwater turtles with five articles per species on average (Table 1). Important variation in research effort across species within a given family was also noticeable (Fig 3).

### Biogeographic, ecological and life history predictors of research effort

Range size was significantly correlated with research effort; species with a larger range attracted more research (using the estimated indigenous (historical) range, n = 370, Spearman ρ = 0.347, p < 0.001; using the current range, n = 313, Spearman ρ = 0.44, p < 0.001). In addition, species that have been successfully introduced outside of their native range had a higher research effort than the other species (Wilcoxon W = 4701.5, p < 0.001; mean research effort = 150.9 ± 48.5 articles for the 58 successfully introduced species, 51.5 ± 16.2 for the other 312 species). Research effort also varied

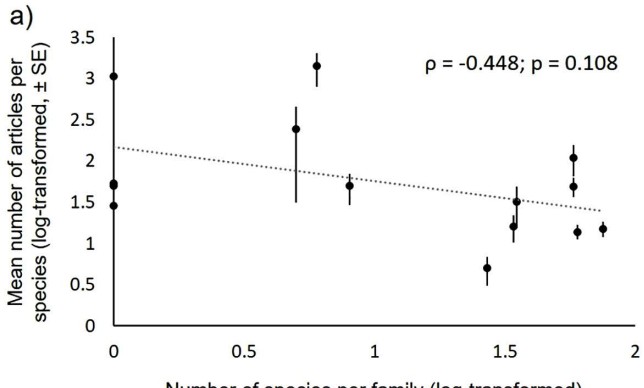

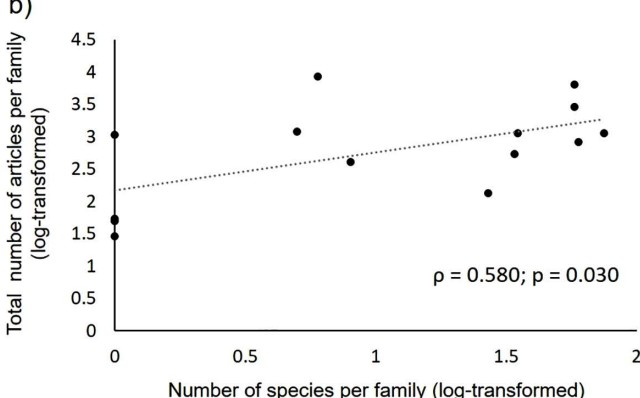

**Fig 2. Association between the number of articles listed in the WoS (either the mean number in panel a, or the total number in panel b) and the number of species per family of Testudines.** The results of Pearson tests are given (ρ = correlation coefficient; p = p-value) for each association.

**Table 1. Species richness, mean research effort per species and total research effort per family in Testudines.**

| | Number of species | Mean number of papers | Total number of papers | Standard error |
|---|---|---|---|---|
| CHELONIIDAE | 6 | 1429 | 8575 | 628 |
| DERMOCHELYIDAE | 1 | 1062 | 1062 | |
| CHELYDRIDAE | 5 | 243 | 1213 | 211 |
| EMYDIDAE | 58 | 111 | 6410 | 45 |
| CARETTOCHELYIDAE | 1 | 54 | 54 | |
| PODOCNEMIDIDAE | 8 | 50 | 402 | 21 |
| PLATYSTERNIDAE | 1 | 50 | 50 | |
| TESTUDINIDAE | 58 | 50 | 2874 | 13 |
| TRIONYCHIDAE | 35 | 32 | 1119 | 17 |
| DERMATEMYDIDAE | 1 | 29 | 29 | |
| KINOSTERNIDAE | 34 | 16 | 548 | 6 |
| GEOEMYDIDAE | 75 | 15 | 1138 | 3 |
| CHELIDAE | 60 | 14 | 833 | 3 |
| PELOMEDUSIDAE | 27 | 5 | 135 | 2 |

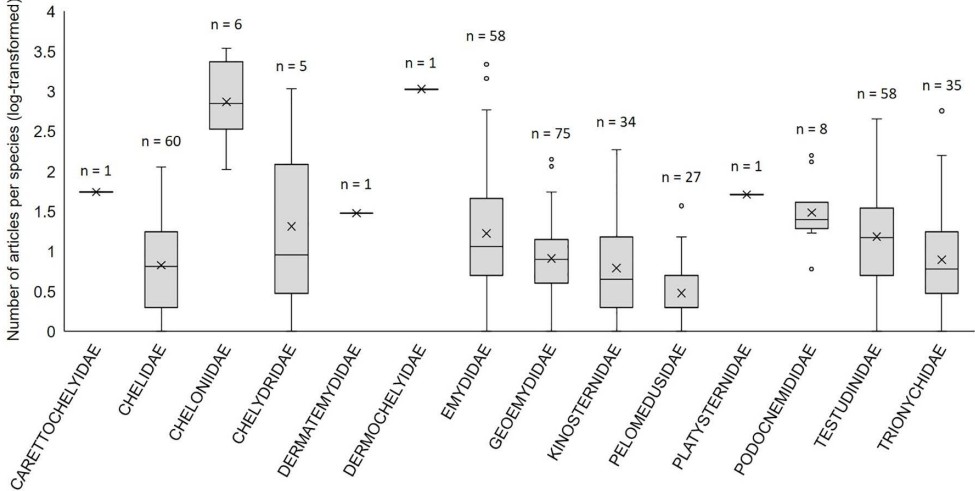

**Fig 3. Box plot of the distribution of the number of articles (log-transformed) per species within each family of Testudines.** Minimum, maximum, median, 25th and 75th quartiles are shown, as well as outliers. The number of species per family is indicated above each box, and x-symbols show the mean for each family.

significantly across geographic realms (Sum sq = 250.18, F = 14.34, Df = 9, p < 0.001). In addition to species occupying multiple realms (mostly species occurring in both Central and North America, or marine species with broad distribution), species from Europe and North America were given more research whereas species from the Caribbean, Central America or Africa were much less studied (see Fig 4). In addition, research effort varied significantly with the research rank of countries where species occur (n = 370, Spearman ρ = 0.436, p < 0.001): research effort was highest for species occurring in at least one of the top 10 countries (n = 152, mean = 145, SE = 37), intermediate for species occurring in at least one of the top 10–20 countries (n = 53, mean = 25, SE = 4), and lowest for species absent from the top 20 countries (n = 165, mean = 9, SE = 1).

 

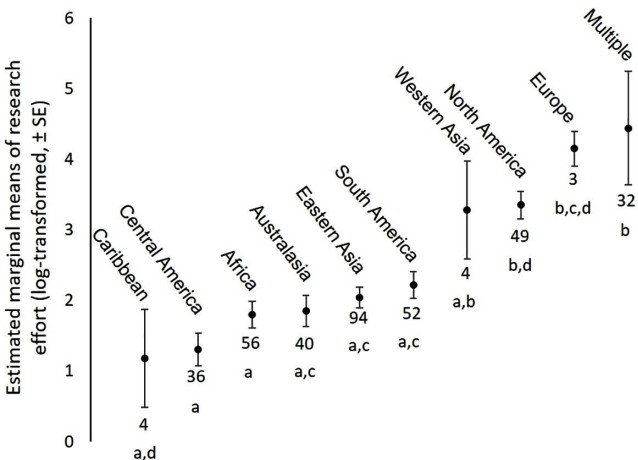

**Fig 4. Variation in research effort across biogeographic realm.** Estimated marginal means were extracted using the emmeans function [53]. Different letters indicate significant differences between geographic realms (e.g., Central America and Europe), whereas research effort does not differ significantly between regions that share the same letter (e.g., all realms with an **a)**. The number of species per realm is indicated.

Research effort did not vary significantly between island and continental species (n = 370, Wilcoxon W = 6076, p = 0.082), but habitat was a significant predictor of research effort (n = 352, Sum sq = 140.81, F = 21.917, Df = 3, p < 0.001; Fig 5), with marine species being studied more frequently than species from any other habitat category (all p < 0.001). Research effort did not differ significantly between freshwater, terrestrial and mixed-habitat species (all p > 0.138). Focusing on either freshwater or terrestrial species, the type of freshwater or terrestrial habitat did not significantly predict variation in research effort (freshwater: n = 126, Sum sq = 5.365, F = 1.243, Df = 2, p = 0.292; terrestrial: n = 54, Sum sq = 5.972, F = 1.573, Df = 2, p = 0.217).

Research effort did not vary across diet categories either (n = 257, Sum sq = 0.89, F = 0.174, Df = 2, p = 0.841). Considering life history traits, research effort increased significantly with body size (maximum SCL: n = 369, Spearman ρ = 0.210, p < 0.001), maximum longevity (n = 202, Spearman ρ = 0.339, p < 0.001) and clutch size (n = 275, Spearman ρ = 0.148, p = 0.014), decreased significantly with egg size (n = 189, Spearman ρ = −0.161, p = 0.026) but was not significantly correlated with generation length (n = 139, Spearman ρ = 0.017, p = 0.839), female age at maturity (n = 92, Spearman ρ = 0.151, p = 0.150) or incubation time (n = 212, Spearman ρ = −0.127, p = 0.065).

Building separate phylogenetic mixed models for each of these 14 predictors led to similar conclusions, except for the insularity effect which became significant – insular species had a significantly lower research effort than continental ones (*p*MCMC = 0.002) – and for incubation time as research effort decreased significantly with incubation time (*p*MCMC = 0.028) once correcting for phylogenetic non-independence (Table 2).

The best model with multiple predictors included five different predictors, in addition to the random effect of phylogeny: range size, introduction status, biogeographic realm, maximum SCL, and egg length. The direction of their effects was similar to those detected using univariate models (see Table 3). Excluding marine species resulted in the same best model and qualitatively similar results (see Suppl. Mat. 1).

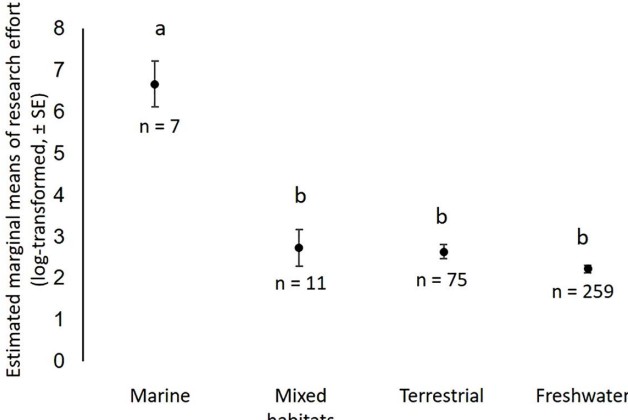

**Fig 5. Variation in turtle research effort across habitats.** Marine species were significantly more studied than turtles from any other habitat category, including species occurring in multiple habitats. The letters a and b indicate the significance (p < 0.05) of between-habitats comparisons (if the difference is significant, the two habitats were attributed different letters; if non-significant, they share the same letter). The number of species per habitat is indicated.

**Table 2. Association between research effort and 14 predictors in Testudines, after taking phylogenetic relationships into account.** Research effort was included as response variable in 14 different PLMM that included phylogeny as random effect and either one of the 14 predictors as fixed effects. For continuous or binary predictors, pMCMC was used to assess significance (a). The positive effect of introduction means that species that were introduced successfully had a higher research effort; the negative effect of insularity means that insular species had a lower research effort than continental ones. For categorical variables, DIC-based comparisons of models including or excluding each categorical predictor are shown (b) as MCMCglmm does not provide unique pMCMC values for these variables (a negative ΔDIC indicates that the predictor improved the fit of the model). SCL = Straight-line Carapace Length; pm = posterior mean; CI = credibility interval, eff samp = effective sample size, n = number of species included. Significant effects are shown in bold.

| Binary or continuous variables | pm | CI | eff samp | pMCMC | n |
|---|---|---|---|---|---|
| **Historic range size (AOO)** | **0.757** | **[0.631; 0.929]** | **1001** | **<0.001** | **275** |
| **Current range size** | **0.864** | **[0.705; 1.005]** | **1001** | **<0.001** | **251** |
| **Introduction** | **1.161** | **[0.748; 1.570]** | **1001** | **<0.001** | **275** |
| **Insularity** | **−1.144** | **[-1.840; -0.447]** | **1001** | **0.002** | **275** |
| **Country research rank** | **0.731** | **[0.548; 0.918]** | **1001** | **<0.001** | **275** |
| **Max SCL** | **0.015** | **[0.005; 0.024]** | **1001** | **0.008** | **274** |
| **Max longevity** | **0.018** | **[0.009; 0.027]** | **909.6** | **<0.001** | **185** |
| Generation length | −0.007 | [-0.033; 0.018] | 1001 | 0.601 | 110 |
| Maternal age at maturity | 0.026 | [-0.312; 0.440] | 1190 | 0.897 | 85 |
| **Clutch size** | **0.028** | **[0.012; 0.042]** | **1001** | **<0.001** | **246** |
| **Egg length** | **−0.028** | **[-0.054; -0.003]** | **1001** | **0.028** | **172** |
| **Incubation time** | **−0.006** | **[-0.011; -0.001]** | **1001** | **0.028** | **194** |
| | | | | | |
| | | | | | |
| Categorical variables | DIC with | DIC without | ΔDIC | n | |
| **Biogeographic realm** | **909** | **949** | **−40** | **275** | |
| Habitat | 950 | 944 | 6 | 273 | |
| Diet | 786 | 783 | 3 | 229 | |

**Table 3. Best multi-predictor model explaining variation in research effort across species of Testudines. Phylogeny was included as random effect, and we used the MCMCglmm procedure. Non-significant predictors excluded from the best model ($pMCMC > 0.01$ or $\Delta DIC > 5$) are shown in italic style below the factors included in the best model. SCL = Straight-line Carapace Length; pm = posterior mean; CI = credibility interval, eff.samp = effective sample size. 115 species were included in this analysis.**

| Explanatory variable | pm | 95% CI | eff.samp | *pMCMC* |
|---|---|---|---|---|
| *Intercept* | 2.269 | [0.474; 4.226] | 1125.8 | 0.018 |
| Historic range size | 0.668 | [0.408; 0.941] | 1205.5 | <0.001 |
| Introduction | 0.463 | [0.025; 0.905] | 1001 | 0.042 |
| Maximum SCL | 0.854 | [0.470; 1.184] | 900.8 | <0.001 |
| Egg length | −0.433 | [-0.734; -0.133] | 1001 | 0.004 |
| Biogeographic realm ($\Delta DIC = -32$) | | | | |
| *Maximum longevity* | *0.184* | *[-0.059; 0.396]* | *1001* | *0.118* |
| *Insularity* | *0.309* | *[-1.240; 1.734]* | *908.9* | *0.691* |
| *Incubation* | *−0.029* | *[-0.331; 0.211]* | *1001* | *0.859* |
| *Country research rank* | *0.035* | *[-0.318; 0.330]* | *1001* | *0.839* |
| *Diet ($\Delta DIC = 8$)* | | | | |
| *Habitat ($\Delta DIC = 27$)* | | | | |

## Extinction risk

IUCN-assessed species had a higher research effort than non-assessed or data-deficient species (estimate = 0.904 ± 0.176, t = 5.123, p < 0.001; mean number of papers for assessed species = 86.2 ± 22.2; for non-assessed species = 14.8 ± 2.6). The research effort of threatened and non-threatened species did not differ significantly (estimate = −0.099 ± 0.214, t = −0.463, p = 0.644), and the risk of extinction did not significantly predict research effort (n = 160, estimate = 0.104 ± 0.070, t = 1.486, p = 0.138). Similarly, research effort did not vary significantly with population trend (n = 177, estimate = 0.114 ± 0.234, t = 0.488, p = 0.626).

## Discussion

Identifying taxonomic biases in research effort can help future research programs to aim at correcting these biases and improve our global understanding of turtles and tortoises. We underline here that Testudines show strong discrepancies in research effort across species, with the overwhelmingly high proportion of research dedicated to a small number of sea turtle species being particularly striking. Phylogeny itself explained 66% of the variance in research effort, also illustrating that some parts of the Testudines tree have been much more researched than others. Although the overall number of publications focused on Testudines has increased substantially over the last decades, discrepancies across species have been maintained so that understudied species remain understudied while more studied species keep attracting more research. Our results also underline that these research-biases are not neutral with regards to species traits – instead, more research is dedicated towards species that share similar biogeographic, ecological and life history traits. Neither high risk nor low risk species were particularly targeted by researchers since conservation status did not predict research effort.

Our analyses underline the high and non-neutral (with regards to species traits) heterogeneity in research effort across turtle species. This heterogeneity showed few changes over the last decades, as illustrated by the minimal decrease in research load discrepancy between the 1980s and 2010s. This is not unique to turtles, and has been described in other taxonomic groups, including birds [19], mammals [21], fishes [22,23] and amphibians [24]. Although our result suggests a need for a reorientation of research efforts and funding towards less studied species, different explanations can help in interpreting it. First, some species are used as model organisms, responding to the need of acquiring in-depth knowledge

of some species to address more advanced biological questions, and leading to outlier-species which attracted the attention of many researchers. *Trachemys scripta* is one such example: the pond slider has been the focus of research addressing questions in a diversity of disciplines, including molecular biology (e.g., [39]), development biology [40], physiology [41], ecology (e.g., [42]) and cognition (e.g., [43]), although understanding the impacts of this species that is invasive in part of its range also contributed to its success with researchers (e.g., see [44]). The relatively small size of this species, its adaptability to captivity and its distribution (in both its native and introduced range) in areas with high scientific capacity made it an ideal model species to study both in the lab and *in situ*.

Second, some species are the target of research because of their sociocultural importance. The focus on charismatic species in conservation has been seen as a way to attract funding and public attention towards conservation issues, although this has been debated [45–48]. From a global knowledge perspective, this focus on some flagship species targeted because of their charisma is more problematic, as it is likely to both concentrate the funding on a small subset of species, and lead to a lack of representativeness of the knowledge on a given taxonomic group such as the Testudines. For example, the particularly high amount of research dedicated to sea turtles is likely mostly explained by their cultural value [49] and by the fascination that they represent for the public, increasing the probability of research funding attribution for this group of turtles. This can be interpreted as a positive outcome: our knowledge of sea turtles is particularly good, especially for species that are difficult to study given their marine lifestyle. In addition, the unique habitat and migratory behaviors of sea turtles also increases the relevance of investigating these species. However, the contrast with non-marine species highlights how research resources are distributed: on average, a sea-turtle species received 33 times more research than a freshwater or terrestrial species. The endangerment of sea turtle species (five are considered at risk of extinction, one is data deficient and one is considered Least Concern) together with their unique use of the marine habitat might be seen as justifications for the high effort put towards sea turtles. However, this is not specific to sea turtles: only 24% of all freshwater and terrestrial species are considered as least concern or near threatened, with the others being either at risk of extinction, not assessed or data deficient. The high proportion of freshwater and terrestrial species which risk of extinction has not been evaluated (109 species) is also a major concern, as under-studied species tend to have a higher risk of extinction than species that have been evaluated [50,51]. Using sea-turtles as flagship species to attract public attention and funding towards other groups of turtles could be used as a strategy to fill some of these major knowledge gaps.

The ease of researcher access to a given species is also an important factor determining species knowledge production. This is likely reflected in the biogeographic traits that determined research effort in our analyses. Species with a larger range are more likely to include research facilities within their range and are therefore more often investigated, while continental species are often more accessible than insular ones (although the insularity effect was only significant when considered as single predictor in a LMM). Similarly, species distributed in areas with a higher scientific capacity, such as North America and Europe or any of the top 10 or top 20 most scientifically productive countries, are more studied than species from areas with more limited research funding such as Africa or Central America. Noticeably, when including both country research rank and biogeographic realm in the same model, only biogeographic realm was retained, likely because the two variables are redundant. This bias related to the scientific capacity within a species range is in line with the literature: for example, [26] found that reptile species whose range overlaps with biodiversity institutions received more research attention. Some of the most important turtle biodiversity hotspots are in areas that are relatively neglected by researchers (Australasia, Africa, South America; [27]). Similar patterns have been detected in other taxonomic groups, including in reptiles [26], birds [19] and sharks [22], underlining that these research biases are not unique to turtles and tortoises. Therefore, conclusions of large-scale studies are likely to mostly apply to well-studied regions such as North America and Europe, but less so to understudied areas such as Central America or Africa. Overall, although targeting species that are easier to access is logical from both a financial and a logistic perspective, orienting research towards understudied areas, localized and/or insular species to correct these biases is highly relevant for turtles as well as for a diversity of other taxonomic groups.

Several life history traits were associated with research effort. These associations might be the result of differences in how easy it is to study species with particular traits. For example, larger species can be easier to detect and study, and often characterized by a longer lifespan, potentially explaining the positive association between research effort and longevity. Alternatively, for some traits, differences in research effort are likely to directly affect the trait estimates. For example, record longevities are more likely to be documented in intensely studied species than in less studied ones, which can lead to biased differences in the estimated species maximum longevities. Associations between research effort and trait values due to biased estimates or trait-based selection of researched species are likely to have consequences for the interpretation of comparative analyses and literature reviews. Such studies are often meant to be representative of entire taxonomic groups. However, they tend to exclude poorly known species and/or consider biased or imprecise trait estimates due to species knowledge discrepancies, making their conclusions less general than expected. Small turtle and tortoise species with small eggs, small clutches and short longevity are, for example, less well known, and therefore likely to be more often excluded from global analyses, biasing any results to a subset of species that is not representative of the entire group. In contrast, maternal age at reproduction and generation length were not associated with research effort. However, these two traits were available for small numbers of species (110 and 86 species, see Table 2), and the lack of significant association might be due to a lack of statistical power. The results of our multivariate models also illustrate this bias. These models focused on the small subset of species with complete trait data (31% of all species, with a mean research effort of 160 articles per species, as compared to 24 articles per species for species with missing trait data), leading to conclusions that were different from the univariate models that considered larger numbers of species. Overall, the importance of these knowledge differences in shaping the results of broad taxonomic scale analyses needs to be acknowledged in order to properly interpret their results.

Apart from the strong research bias towards marine species, ecological traits did not predict variation in research effort across species. When focusing on non-marine species, research effort did not vary according to Testudines habitat or diet. The lack of differences in research effort related to either habitat or diet is likely robust since these two traits were available for most species. It suggests that species with different habitats (with the exception of marine habitats) and diet have been relatively evenly investigated, contrasting with other taxonomic groups where species from some habitats tend to be under-studied. For example, deep-sea shark species were less often investigated than species occurring closer to the surface [22].

Previous works have evidenced the importance of other traits as predictors of variation in research effort across species. Socioeconomic factors, including range overlap with biodiversity institutions, were for example shown to increase the amount of research dedicated to reptile species [26], while recently described species tend to have received less research [26]. Aesthetic biases have also been demonstrated, for example in plants where more colorful and conspicuous species attracted more research [25]. The use of anthropic habitats also tends to increase research attention, including in amphibians [24]. Such parameters may also influence research effort in turtles and tortoises and could be considered in future studies.

The number of publications per species did not vary with either extinction risk or population trend across Testudines species. Removing sea turtles from the analyses did not change this result (supplementary results S1). Similarly, considering a categorical variable instead of a continuous one to represent either extinction risk or population trend led to the same conclusion. This result contrasts with several other taxonomic groups, where researchers tend to focus more on low risk species (e.g., in amphibians [24] and birds [19]) or species with intermediate risk levels (e.g., in sharks, [22]). This relatively even distribution of research attention across extinction risk categories may be considered as a positive result, especially as compared to other taxa: endangered species are not relatively understudied. However, a special research focus on endangered species would be desirable to improve conservation practices on species that need it the most. A study focusing on research effort on turtles and tortoises in conservation science (rather than all publications) and its distribution according to species risk of extinction would be of interest to better assess the needs for conservation research in endangered species (see for example this study on primates: [21]). The (expected) much lower number of publications on the 110 species that were either not assessed or considered data deficient (as compared to species which extinction risk

has been assessed) also raises important concerns, as these species tend to be at a higher risk of extinction according to work conducted on other taxonomic groups [50,51]. Overall, species that are at high risk of extinction, as well as species which extinction risk has not been assessed yet, or whose extinction risk could not be assessed because of data deficiency, should be the focus of more research to optimize our chances of implementing appropriate conservation actions for the entire clade. Future work investigating the association between research effort and conservation funding per species would also provide valuable information on whether indices of research efforts are also indicative of conservation funding.

Assessing the amount of research dedicated to a given species requires to make choices with regards to the sources and criteria considered. Including one or several databases, considering or not the grey literature, accounting for name changes or quantifying the amount of research dedicated to a given species within publications are among the important factors that could affect conclusions. The index we are using cannot be directly interpreted as an absolute measure of the global research effort that was dedicated to a given species. Studies published before 1975 were not included in this index, and grey literature as well studies not included in the WoS were excluded when calculating this index. In addition, even when ignoring these limitations, the number of articles mentioning a species name in its title or abstract is a rough proxy for research effort: the amount of published research and knowledge provided on a given species can vary substantially from one article to another, and using the number of articles is an important shortcut. Other indices, including information on species knowledge (e.g., trait value knowledge, number of individuals measured, duration of studies) would provide complementary information on how much research effort has been invested in a given species. However, these limitations are mostly important when aiming at obtaining absolute indices able to capture all or most research that has been conducted on a given species. Here, our purpose was to obtain a metric that allows comparisons of research effort across species, i.e., to evaluate the amount of research attributed to a given species relative to others. To that aim, we demonstrated that the choice of one index over another had little importance for Testudines, since all tested indices were highly correlated. Including synonyms or not, considering all fields or only the titles during the search for papers, and considering the Web of Science or Scopus databases resulted in comparable numbers of publications per species (all Pearson coefficients > 0.94) – a pattern similar to other taxonomic groups [19,22]. That being said, we recommend using the index that takes into account synonyms (see [29]) and all fields (rather than focusing on articles where the species names appears in the title), as it is more inclusive.

By identifying both the most and least studied families and species of Testudines, we hope that this work will help orienting research towards taxa and regions that require more research. Families such as the Pelomedusidae, with only five articles per species on average (Table 1), would, for example, warrant additional attention. Although having model species is relevant and allows addressing fundamental questions in more depth, the lack of knowledge on a significant proportion of the Testudines is likely to affect our interpretation of their global ecological importance and our ability to preserve them. More research on endangered species, including in areas such as Southeast Asia that concentrate a high proportion of high risk or unassessed species [52], but also to understudied areas such as Africa and Central America, would be especially rewarding by both increasing our knowledge of this unique taxonomic group and identifying actions that could help improve the conservation status of endangered species. Sharing funding allocated to sea turtle research and conservation might also be a way forward to fill the knowledge gaps that are likely to prevent effective conservation actions for terrestrial and freshwater species that also play key roles in their ecosystems. By providing an index of research effort for all species of Testudines, we also hope to help account for potential biases related to discrepancies in research effort across species in global reviews and comparative analyses.

## Supporting information

**S1 File. Best multi-predictor model explaining variation in research effort across species of Testudines after excluding marine species.** Phylogeny was included as random effect, and we used the MCMCglmm procedure. Non-significant predictors excluded from the best model ($p$MCMC > 0.01 or $\Delta$DIC > 5) are shown in italic style below the

factors included in the best model. SCL = Straight-line Carapace Length; pm = posterior mean; CI = credibility interval, eff. samp = effective sample size. 110 species were included in this analysis.
(DOCX)

**S2 File. Dataset including the research effort indices and traits considered in the analyses.**
(XLSX)

**S3 File. R code used for the conducted analyses.**
(R)

**S4 File. List of the 32 species with no publication record in the Web Of Science between 1975 and 2023.**
(DOCX)

## Acknowledgments

We are grateful to Tomasz Szczygielski and an anonymous reviewer for comments on a previous version of the manuscript.

## Author contributions

**Conceptualization:** Simon Ducatez, Jayna Lynn DeVore.

**Data curation:** Simon Ducatez.

**Formal analysis:** Simon Ducatez.

**Funding acquisition:** Simon Ducatez, Jayna Lynn DeVore.

**Investigation:** Simon Ducatez, Jayna Lynn DeVore.

**Methodology:** Simon Ducatez, Jayna Lynn DeVore.

**Project administration:** Simon Ducatez.

**Visualization:** Simon Ducatez.

**Writing – original draft:** Simon Ducatez.

**Writing – review & editing:** Simon Ducatez, Jayna Lynn DeVore.

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
