## [Decision Letter · Decision Letter 0]

12 Sep 2025

PONE-D-25-36542Which turtle should I study? Uneven distribution of research effort across Testudines speciesPLOS ONE?

Dear Dr. Ducatez,

Thank you for submitting your manuscript to PLOS ONE. After careful consideration, we feel that it has merit but does not fully meet PLOS ONE’s publication criteria as it currently stands. Therefore, we invite you to submit a revised version of the manuscript that addresses the points raised during the review process.

However, after carefully reviewing the manuscript and considering the feedback provided by the selected reviewers, I recommend that this paper can be considered for publication after the author (s) undertake major revisions.

The key points that I would like to highlight are as follows:

**Methodology**: The study relies on the number of articles retrieved from the Web of Science. It is understood that a manual review of all relevant publications is impossible; however, semi-automated search strategies (WOS) may risk missing relevant data or including irrelevant ones. These issues should be more explicitly acknowledged. And discussed.: The study relies on the number of articles retrieved from the Web of Science. It is understood that a manual review of all relevant publications is impossible; however, semi-automated search strategies (WOS) may risk missing relevant data or including irrelevant ones. These issues should be more explicitly acknowledged. And discussed.: The study relies on the number of articles retrieved from the Web of Science. It is understood that a manual review of all relevant publications is impossible; however, semi-automated search strategies (WOS) may risk missing relevant data or including irrelevant ones. These issues should be more explicitly acknowledged. And discussed.: The study relies on the number of articles retrieved from the Web of Science. It is understood that a manual review of all relevant publications is impossible; however, semi-automated search strategies (WOS) may risk missing relevant data or including irrelevant ones. These issues should be more explicitly acknowledged. And discussed.**Geopolitical bias**: Research effort likely reflects whether species occur in scientifically active countries (e.g., USA, Europe, and China). This should be analyzed more directly rather than only by broad regions.: Research effort likely reflects whether species occur in scientifically active countries (e.g., USA, Europe, and China). This should be analyzed more directly rather than only by broad regions.: Research effort likely reflects whether species occur in scientifically active countries (e.g., USA, Europe, and China). This should be analyzed more directly rather than only by broad regions.: Research effort likely reflects whether species occur in scientifically active countries (e.g., USA, Europe, and China). This should be analyzed more directly rather than only by broad regions.**Taxonomy update**: Acknowledge the new 2025 Turtles of the World checklist (359 turtle and tortoise species now face extinction vs. 357 in 2021).: Acknowledge the new 2025 Turtles of the World checklist (359 turtle and tortoise species now face extinction vs. 357 in 2021).: Acknowledge the new 2025 Turtles of the World checklist (359 turtle and tortoise species now face extinction vs. 357 in 2021).: Acknowledge the new 2025 Turtles of the World checklist (359 turtle and tortoise species now face extinction vs. 357 in 2021).

For more information, please see the comments from reviewers.

We look forward to receiving your revised manuscript.

Kind regards,

Amaal Gh. Yasser, Ph.D.

Academic Editor

PLOS ONE

“Agence nationale de la recherche, France InvEcoF #ANR-22-CE02-0007

Fondation Fyssen, France.”

“SD and JD were funded by the Agence nationale de la recherche, France InvEcoF #ANR-22-CE02-0007 and the Fondation Fyssen, France.”

“Agence nationale de la recherche, France InvEcoF #ANR-22-CE02-0007

Fondation Fyssen, France.”

Additional Editor Comments:

Dear authors,

After carefully reviewing the manuscript and considering the feedback provided by the selected reviewers, I recommend that this paper can be considered for publication after the author (s) undertake major revisions.

For more information, please see the comments from reviewers.

Reviewers' comments:

Reviewer's Responses to Questions

**Comments to the Author**

1. Is the manuscript technically sound, and do the data support the conclusions?

Reviewer #1: Yes

Reviewer #2: Yes

2. Has the statistical analysis been performed appropriately and rigorously?

Reviewer #1: No

Reviewer #2: I Don't Know

3. Have the authors made all data underlying the findings in their manuscript fully available?

Reviewer #1: Yes

Reviewer #2: Yes

4. Is the manuscript presented in an intelligible fashion and written in standard English?

Reviewer #1: Yes

Reviewer #2: Yes

Reviewer #1: This manuscript addresses a timely and important issue uneven research effort across Testudines species, highlighting biases and gaps that affect both ecological understanding and conservation actions. The study is ambitious in scale, data-rich, and well-documented, but several concerns need to be addressed before it can be considered for publication in PLOS ONE:

1. Justification of Methodological Choices (Search Strategy & Data Limitations)

The authors rely heavily on the number of articles retrieved from Web of Science using species names. While they attempt to validate the approach through correlation with Scopus data and synonym comparisons, this method can still under- or over-estimate research effort (e.g., name ambiguity, short genus names). Consider a more detailed justification or a sensitivity analysis beyond correlation coefficients.

2. Treatment of Sea Turtles as Outliers

While the rationale to exclude sea turtles from the multivariate analysis is understandable due to their high representation, this introduces inconsistency: the main claim about marine species dominating research becomes methodologically decoupled from the model-based inference. Consider conducting parallel analyses (with and without sea turtles) to better understand their effect.

3. Insufficient Integration of Conservation Implications

The Discussion does mention that endangered species are not more studied, but the conservation implications remain superficial. The authors should provide clearer suggestions: How can their findings be translated into priority research planning, especially for data-deficient taxa?

4. Trait–Effort Relationships Need Clarification

The final multivariate model includes biologically meaningful traits like body size, longevity, and clutch size. However, clutch size switches from a positive association in the univariate to negative in the multivariate model, without adequate explanation. Please address potential collinearity or scaling issues, and interpret biological significance more clearly.

5. Geographic Bias Discussion Needs Expansion

The paper identifies geographic disparities (e.g., North America vs. Africa), but lacks explanation of underlying drivers (e.g., funding, infrastructure, language bias). Incorporate literature from bibliometric studies or conservation research bias.

Suggested Additional Analyses (if feasible):

1- A network map of author affiliations vs. species studied might add depth to understanding geographic biases.

2- A comparison between publication count and conservation funding per species or region, if data are available.

Conclusion:

This manuscript has the potential to make a meaningful contribution to reptile conservation and bibliometric ecology, but it requires significant revisions to improve its methodological rigor, analytical clarity, and conservation relevance.

Please revise the manuscript accordingly and provide a detailed point-by-point rebuttal.

Reviewer #2: This is an interesting contribution with implications not only for the turtles, but for the approach to biodiversity studies as a whole. I need to stress that I am not an expert in statistics, so those should be checked by a specialist in the field. However, I can provide some insights from the general biological and herpetological point of view. Overall, I think the paper is well-written and it should be published but I see some issues, mostly with how the results are presented, as I think that the potential of the data is not fully utilized and some aspects may require slight refinement or a better justification.

The topic is somewhat difficult from the methodological standpoint because, given the vast number of published papers, it is not really possible to review their contents manually, whereas any semiautomatic approach (such as the one employed by the Authors) will unavoidably result in some errors. For example, some papers providing valuable information on some species may be overlooked, because they do not include the full name of the species in any of the fields searchable by the Web of Science, but instead use, e.g., only the generic name or the name of the family (as I understand, the Authors did not look for those, only for the binomina). On the other hand, in some cases a specific name may be used in an abstract, e.g., solely in a comparative context without any meaningful information on that species presented in the paper (false hits), or as a synonym (double hits when both variants of a name are searched for). I understand that these instances are inevitable and while some workarounds can be proposed, they cannot be fully eliminated. I also understand that the cutoff used by the Authors (1975) is a sad necessity due to the increasingly limited availability of data for processing (that could be, to some extent, helped by inclusion of, e.g., the Biodiversity Heritage Library) but also due to complications regarding the synonymy and more frequent usage of sole generic names in those older papers. Unfortunately, this leaves out some important anatomical and ecological literature, also regarding the less popular taxa, which is still useful and informative and which in some cases may render restudy of those species in modern times less of a priority, thus causing them to appear even less studied in the selected timeframe. This should be perhaps acknowledged slightly more in the text (although the Authors already referred to that problem) but other than that we cannot do much about it and I accept that as an inherent limitation of that kind of research.

One oversight and potential problem I see with communicating the methodology is related to Fig. 1. It presents the number of review papers on turtles. Unfortunately, I don’t see any notion of such a search in the Methods section and the methodology for compiling that charts seems to be only explained in the figure caption. However, this does not seem right for two reasons. (1) using solely the given query (“tortoise*” OR “chelonian*” OR “turtle*” OR “testudines*”) one is expected to find not only papers relevant to the reviewed manuscript, but potentially also papers completely out of scope, e.g., on extinct stem turtles, turtle-derived materials (tortoiseshell), arthropods (tortoise beetles), etc., so the result does not really say much. (2) I tried to see how badly out of scope the results are in that case, but I could not replicate the Author’s results. I searched for the provided keywords, string copied directly from the figure caption, and document type set to Review Article. I will give the values for the year 2022, because this is the largest number in the Author’s search and shows the problem the most easily, but there were also differences for other years. When searching in Author Keywords, 21 articles were found. This is much below the over 25 papers found by the Authors for that year. When searching in Keywords plus, the search yielded 58 papers for that year – this is much more. Also, as expected, the search found lots of papers on fossil turtles and some nonsensical ones (e.g., “Plant phenolics: neglected secondary metabolites in plant stress tolerance”). Please check and preferably refine – the Author Keywords yielded more sensible results in that case and about 250 papers total, so it is doable to leave out those that are out of scope). In case the Authors searched in Keywords Plus and manually rejected results out of scope (possible but I think unlikely, as that search produced over 700 results for me) then kudos to them but please communicate that in the Methods!

The most important aspect, which is currently very indirectly and tangentially tackled by the Authors but certainly requires more attention as a potential good predictor of research effort, is simply whether the species in question is native to territories of more scientifically active countries (e.g., with better financing of scientific institutions). This may very well explain why some taxa became sort of model species for turtles and why they continue to gain attention while other remain neglected. Given the huge differences in research effort between species, some receiving next to no attention and some literally thousands of papers, I suspect that such geopolitical factors may not only be significant, but even overpower other traits – especially that, if I see correctly, all but one species with over 150 papers on their account occur either in the USA, Western or Southern Europe, or China, whereas none of at least several dozens of the least studied species (except of some artificially underestimated instances – see below) occurs there. Currently, the Authors only briefly consider that topic in terms of generalized geographic regions (North America, Central America, South America, etc.) but I am afraid this misses a lot of nuance, especially that a number of taxa falls into the uninformative “Multiple” category. Analyzing whether there is a statistically significant correlation between the research interest and the overlap of the species range with the territory of one or more of, let’s say, the 10 or 20 most scientifically active countries may prove extremally interesting and should not require a lot of additional work and may explain why some taxa inhabiting the same large-scale geographic area differ significantly in attention. I propose that the Authors check and discuss whether this is indeed the case.

Additionally, the Authors should address the problem of recently established species, such as Kinosternon steindachneri (2013) and Sternotherus intermedius (2017), which did not yet accumulate a significant bibliography of their own, but were considered historically as part (subspecies or subpopulation) of their closely related species (in the case of Kinosternon steindachneri – Kinosternon subrubum) or hybrids (in the case of Sternotherus intermedius). As a result, true research effort devoted to them is either difficult to establish or artificially underestimated, in turn possibly skewing indices for corresponding supraspecific taxa and traits. If possible, in those instances it would be advisable to track the research effort on the subspecies rather than species level. Otherwise, they should probably be either considered jointly with their sister species or left out, unless their impact is confirmed to be insignificant.

Another thing that may deserve reconsideration on the part of the Authors is the usage of full decades in some of their analyses, despite their timeframe starting and ending in the middle of the decades (1975–2024). Five-year bins may perhaps be more adequate?

Finally, the paper is mostly built on the taxonomical framework established by the 2021 ninth edition of the Turtles of the World checklist. However, a new edition (tenth, 2025) is already available, so the manuscript should be updated to acknowledge that. The new edition distinguishes 359 extant species instead of 357 recognized by TotW 2021. Whether those new species should be considered on their own depends on the approach (see above), but a lot of attention has been given to turtle extinctions in the new volume, so it is relevant at least in that context.

More detailed notes are provided below:

P. 3, lines 43–44 (“Testudines emerged during the Triassic and survived two mass extinction crises that eliminated several other groups of reptiles at the end of the Permian and Cretaceous”): This statement is incorrect for two reasons. First, the Testudines are the crown group composed from two extant clades: the side-necked turtles (Pleurodira) and hidden-necked turtles (Cryptodira). The exact time of divergence or those two clades (and, in consequence, the appearance of the Testudines) is somewhat contentious, as it depends on the classification of some poorly known fossil taxa, but it was certainly later than in the Triassic, most probably in the Middle to Late Jurassic. All Triassic turtles are stem taxa so they belong to Testudinata, not Testudines. Second, the Permian was before the Triassic, so neither the Testudines nor the Testudinata (as a group appearing at the end of the Triassic) could not survive the Permian extinction. I imagine that the Authors may confuse here the Testudines, Testudinata, and Pan-Testudines/Pantestudinata (a larger group including also shell-less turtle relatives, possibly also the Permian Eunotosaurus africanus), but these not the same and the paper focuses solely on Testudines proper, so there is no need to bring those earlier forms up.

P. 3, lines 48 and 87 (“Squamates”, “Amphibians”): when anglicized, the name should not be capitalized (so “Squamata” and “Amphibia” would be correct but in that case “squamates” and “amphibians”).

P. 4, lines 104–105 (“fast lifestyle”, “slow lifestyle”): this seem somewhat poorly fitting terms, maybe “life cycle”?

P. 6, line 186 (“maximum standard carapace length”): I am unfamiliar with such a measurement and I think the Authors mean here the “maximum straight-line carapace length” (a common measurement used for turtles and typically abbreviated as SCL, also used by Rhodin et al., 2021). If so, please also correct elsewhere. If not, please define.

P. 8, line 313: can this increase be partially explained by the incomplete sampling of older articles in the WoS, e.g., published in historical but now discontinued journals which are not indexed?

P. 10, line 401 (“Trachemys scrypta”): Trachemys scripta.

Table 1: I think I understand what the Authors want to show here, but at the same time there is something missing. The juxtaposition of the taxonomic richness of the family and the number of papers is some information, but this does not say too much without some measure of the evenness of coverage. As presented, the Chelydridae are in the third spot with a decent 243 papers per species, which looks very good, but what of it, if nearly 90% of that (!) is devoted to one species. Conversely, the Geoemydidae only have 16 papers per species, so they appear significantly understudied compared to the chelydrids, but in reality the research effort is much more evenly distributed. In consequence, I feel the message here falls into the same pitfall which the Authors try to solve in their manuscript – we see an averaged picture but completely miss how some species are understudied. Also, the numbers alone are difficult to visualize, so the nice data that the Authors gathered get somewhat lost. I would strongly suggest presenting this in some more visual way. Perhaps a bar chart with the bars subdivided into sections corresponding to the proportional research attention received by individual species, or a pie chart with subdivided slices? I get it that the number of species within many families is too large to show all of them within a single bar/slice, but the differences in attention between them are large enough that the best studied species within each family should be visible as individual portions, whereas the least studied species may be grouped together. This would not only nicely show the differences between the families, but it would also present the disproportions in the effort within each of them and it would also be much more visually appealing.

Fig. 1 caption: “Evolution” is not a good term here, I would suggest just leaving “The number of review articles” etc.

Figs 4 and 5: Please explain what do “a”, “b”, and “c” mean.

.

Reviewer #1: No

Reviewer #2: **Yes:**Tomasz SzczygielskiTomasz SzczygielskiTomasz SzczygielskiTomasz Szczygielski

---

## [Author Response · Author response to Decision Letter 1]

27 Nov 2025

The key points that I would like to highlight are as follows:

1. Methodology: The study relies on the number of articles retrieved from the Web of Science. It is understood that a manual review of all relevant publications is impossible; however, semi-automated search strategies (WOS) may risk missing relevant data or including irrelevant ones. These issues should be more explicitly acknowledged. And discussed.

Our answer : We now discuss these issues more explicitly in both the methods and discussion (see lines 139 to 164 and 531 to 555). We have now developed this part of the discussion in more details to clearly underline the limitations of this index of research effort and suggest others that could give complementary information. However, as mentioned in our responses to the reviewers’ comments, we believe that this is a limited issue, as our objective was not to assess the absolute research effort, but to compare research effort across species. For example, we state when describing our methods: “Note that these indices are proxies for research effort, and not absolute estimates. They, for example, ignore the work that was published before 1975 as well as any reference that is not referenced in the WoS, including references from the grey literature. However, by constructing these indices, our purpose was to obtain a relative metric of research effort to compare species with each other, rather than an exact number of articles published per species. To test whether using different databases could affect this relative research effort score, we also compared the index obtained using the WoS database with the index extracted by Guedes et al. (2023) using Scopus and available for 343 of the 370 species considered.”

2. Geopolitical bias: Research effort likely reflects whether species occur in scientifically active countries (e.g., USA, Europe, and China). This should be analyzed more directly rather than only by broad regions.

Our answer: We now included an extra variable considering each species presence/absence in the 10 and 20 most scientifically active countries. It is indeed a significant predictor, though it did not affect our conclusions with regards to the other predictors, and it was not retained in our multi-predictors model.

3. Taxonomy update: Acknowledge the new 2025 Turtles of the World checklist (359 turtle and tortoise species now face extinction vs. 357 in 2021).

Our answer: We now updated the taxonomic list to the 2025 checklist, re-running all analyses after including the extra species. Note that this update was published in July 2025, after we submitted the first version of our manuscript.

Our answer: We have added our code to the Suppl. Mat.

“Agence nationale de la recherche, France InvEcoF #ANR-22-CE02-0007

Fondation Fyssen, France.”

Our answer: Changed accordingly

“SD and JD were funded by the Agence nationale de la recherche, France InvEcoF #ANR-22-CE02-0007 and the Fondation Fyssen, France.”

“Agence nationale de la recherche, France InvEcoF #ANR-22-CE02-0007

Fondation Fyssen, France.”

Our answer: We have now removed the funding information from the acknowledgements section and included our funding statement and role of funder statement in the cover letter.

Our answer: We have now included the captions for Supporting Information at the end of our manuscript.

Additional Editor Comments:

Dear authors,

After carefully reviewing the manuscript and considering the feedback provided by the selected reviewers, I recommend that this paper can be considered for publication after the author (s) undertake major revisions.

For more information, please see the comments from reviewers.

Reviewers' comments:

Reviewer's Responses to Questions

Comments to the Author

Review Comments to the Author

Reviewer #1: This manuscript addresses a timely and important issue uneven research effort across Testudines species, highlighting biases and gaps that affect both ecological understanding and conservation actions. The study is ambitious in scale, data-rich, and well-documented, but several concerns need to be addressed before it can be considered for publication in PLOS ONE:

Our answer: We thank the reviewer for comments that helped improving the manuscript.

1. Justification of Methodological Choices (Search Strategy & Data Limitations)

The authors rely heavily on the number of articles retrieved from Web of Science using species names. While they attempt to validate the approach through correlation with Scopus data and synonym comparisons, this method can still under- or over-estimate research effort (e.g., name ambiguity, short genus names). Consider a more detailed justification or a sensitivity analysis beyond correlation coefficients.

Our answer: We further developed this important point in the discussion (see lines 535-547). Note that our index of research effort is not meant to provide an absolute, but a relative metric for research effort, as mentioned in the methods section. The objective is to compare species, not to know exactly how much research was devoted to each specie. As a result, the number of articles on large online databases such as Scopus and Web of Science have been largely used in previous work investigating research effort in other taxonomic groups. We already provide multiple sensitivity analyses, considering different search strategies in the search engines (title only, or more global search), multiple search engines (Scopus or Web of Science) and considering synonyms or not. Although other sensitivity analyses could be considered, they would rely on extracting information from the grey literature or an in-depth analysis of each of the thousands of published articles that focus on turtles. We are not competent to automatize this type of analysis at this stage, and do not believe that it would bring much more information than confirming, with yet an extra sensitivity analysis, that all metrics for research effort are highly correlated. Note also that we are using an index that has been used by multiple authors in previously published work (including reptile work), and we have included more tests of the robustness of our index than most published work investigating taxonomic variation in research effort. This approach is a standard way to assess research effort and has been used by several papers assessing the heterogeneous study effort across species and its drivers (e.g., to illustrate how frequently these indices are used, here are a few examples of articles using similar indices, in addition to the articles already cited in our manuscript).

Adamo M, Chialva M, Calevo J et al (2021) Plant scientists’ research attention is skewed towards colourful, conspicuous and broadly distributed flowers. Nat Plants 7:574–578.

Mammola S, Adamo M, Antić D et al (2023) Drivers of species knowledge across the tree of life. eLife 12:RP88251

Simoncini, A., Ficetola, G.F. & Lattanzi, M. Taxonomic bias towards charismatic and easy-to-find mammals shapes knowledge of parasites. Biodivers Conserv (2025). https://doi.org/10.1007/s10531-025-03147-1

Tam J, Lagisz M, Cornwell W et al (2022) Quantifying research interests in 7,521 mammalian species with h-index: a case study. Gigascience 11:giac074.

Wilson JRU, Proches S, Braschler B et al (2007) The (bio)diversity of science reflects the interests of society. Front Ecol Environ 5:409–414. https://doi.org/10.1890/060077.1

2. Treatment of Sea Turtles as Outliers

While the rationale to exclude sea turtles from the multivariate analysis is understandable due to their high representation, this introduces inconsistency: the main claim about marine species dominating research becomes methodologically decoupled from the model-based inference. Consider conducting parallel analyses (with and without sea turtles) to better understand their effect.

Our answer: We agree with the reviewer that parallel analyses including vs excluding sea turtles for the global model would be preferable. We now ran the multivariate analysis twice, either with or without sea turtles. We show the results of the model on all species in the main text, and the results of the model excluding sea turtles in the Supplementary Material. Note that the results were consistent between these two datasets.

3. Insufficient Integration of Conservation Implications

The Discussion does mention that endangered species are not more studied, but the conservation implications remain superficial. The authors should provide clearer suggestions: How can their findings be translated into priority research planning, especially for data-deficient taxa?

Our answer: We added more discussion on this aspect lines 524-530. Note that our analyses mostly identify a need for more work to clearly identify how these research gaps can affect our ability to design conservation plans for endangered species, as discussed in the paragraph lines 507-530. Specifically, our findings underline that species that are at a higher risk of extinction, though they should be prioritized by researchers to favor their conservation and/or recovery, are not more often studied than less threatened species. We believe that our index or research effort can be used as an extra argument to help obtaining funding for under researched species (either data deficient or unassessed ones) and endangered ones.

4. Trait–Effort Relationships Need Clarification

The final multivariate model includes biologically meaningful traits like body size, longevity, and clutch size. However, clutch size switches from a positive association in the univariate to negative in the multivariate model, without adequate explanation. Please address potential collinearity or scaling issues, and interpret biological significance more clearly.

Our answer: We agree with the reviewer that this switch is unclear and could be due to collinearity issues, or differences in the set of species considered in the monovariate and multivariate models. The correlation between max SCL and clutch size was of 0.568 – to be safe and avoid any risk of collinearity, we therefore decided to exclude clutch size from the multivariate model.

Note that to better understand the switch in clutch size effect (even though we are not presenting this switch in the MS anymore), we built a model with clutch size as the only predictor considering the species that were included in the multivariate model only, to determine whether the switch could be due to a change in the set of species considered. This resulted in a non-significant effect of clutch size, supporting the idea that both collinearity between clutch size and SCL, and the subsample of species on which the multivariate analysis could be conducted, explain the discrepancy between the univariate and multivariate models with regard to the clutch size effect.

5. Geographic Bias Discussion Needs Expansion

The paper identifies geographic disparities (e.g., North America vs. Africa), but lacks explanation of underlying drivers (e.g., funding, infrastructure, language bias). Incorporate literature from bibliometric studies or conservation research bias.

Our answer: We have now added an extra analysis as suggested by reviewer 2, comparing species present in the 10 or 20 most scientifically active countries with species that are absent from these countries. We also added sentences of discussion on this aspect lines 459-461. This was also discussed in the paragraph lines 499-506.

Suggested Additional Analyses (if feasible):

1- A network map of author affiliations vs. species studied might add depth to understanding geographic biases.

Our answer: Although we agree that this would be an interesting addition, this was out of the scope of the manuscript and would require data extraction from each of the thousands of articles on the different turtle species. All of these data extractions were conducted by hand, so this would add a massive amount of work. A future study using AI might bring this information more easily.

2- A comparison between publication count and conservation funding per species or region, if data are available.

Our answer: We also agree that this would be a welcome future study, though this was out of the scope of our study. We thank the reviewer for this suggestion which we added as a perspective in the discussion (see lines 528-530).

Conclusion:

This manuscript has the potential to make a meaningful contribution to reptile conservation and bibliometric ecology, but it requires significant revisions to improve its methodological rigor, analytical clarity, and conservation relevance.

Please revise the manuscript accordingly and provide a detailed point-by-point rebuttal.

Our answer: We are grateful to the reviewer for constructive comments that helped us improving the quality of our study.

Reviewer #2: This is an interesting contribution with implications not only for the turtles, but for the approach to biodiversity studies as a whole. I need to stress that I am not an expert in statistics, so those should be checked by a specialist in the field. However, I can provide some insights from the general biological and herpetological point of view. Overall, I think the paper is well-written and it should be published but I see some issues, mostly with how the results are presented, as I think that the potential of the data is not fully utilized and some aspects may require slight refinement or a better justification.

Our answer: We thank the reviewer for comments that helped improving the manuscript.

The topic i

---

## [Decision Letter · Decision Letter 1]

19 Feb 2026

PONE-D-25-36542R1Which turtle should I study? Uneven distribution of research effort across Testudines speciesPLOS One?

Dear Dr. Ducatez,

Thank you for submitting your manuscript to PLOS ONE. After careful consideration, we feel that it has merit but does not fully meet PLOS ONE’s publication criteria as it currently stands. Therefore, we invite you to submit a revised version of the manuscript that addresses the points raised during the review process.

This manuscript is greatly improved compared to the original submission, and is near ready for publication.  However, reviewer two has identified some additional minor revisions that could further enhance the clarity of the manuscript.

We look forward to receiving your revised manuscript.

Kind regards,

Patrick R Stephens, Ph.D.

Academic Editor

PLOS One

Journal Requirements:

Reviewers' comments:

Reviewer's Responses to Questions

**Comments to the Author**

Reviewer #3: All comments have been addressed

Reviewer #4: All comments have been addressed

2. Is the manuscript technically sound, and do the data support the conclusions?

Reviewer #3: Yes

Reviewer #4: Yes

3. Has the statistical analysis been performed appropriately and rigorously?

Reviewer #3: Yes

Reviewer #4: Yes

4. Have the authors made all data underlying the findings in their manuscript fully available?

Reviewer #3: Yes

Reviewer #4: Yes

5. Is the manuscript presented in an intelligible fashion and written in standard English?

Reviewer #3: Yes

Reviewer #4: Yes

Reviewer #3: The revised manuscript was checked and there is no error found in the text. I checked the comments and the author responses. It is sufficient for publication now.

Reviewer #4: The authors have done commendable work in revising this manuscript. The revisions are substantial and demonstrate a genuine engagement with the concerns raised by both reviewers and the editor. In particular, the updated taxonomy (2025 Turtles of the World checklist, now covering 370 species), the addition of the country research rank variable to address geopolitical bias, the parallel multivariate analyses with and without sea turtles, and the resolution of the clutch size sign reversal through its exclusion from the multivariate model are all welcome improvements. The removal of the problematic review papers figure and the correction of the Triassic/Testudines error also strengthen the manuscript. The paper reads more clearly and is better supported than the first submission.

That said, a small number of issues remain, which I outline below. Most are minor and should be straightforward to address. I believe the manuscript is approaching publication readiness, and I would recommend minor revisions before acceptance.

Substantiative points

1. Wording error on line 326 (persists from the previous version). The sentence currently reads: "This increase was particularly important in marine turtles, which attracted 18.1 times more research in the 1980s as compared to the 2010s." The direction is reversed. Given the context and Figure 1, the intended meaning is clearly that the 2010s saw 18.1 times more research on marine turtles than the 1980s, not the other way round. This should be corrected to read something like: "...which attracted 18.1 times more research in the 2010s as compared to the 1980s."

2. Potential bias in the multivariate model subset. The final multi-predictor model (Table 3) is based on 115 species for which all trait values were available, out of 370 total. This represents roughly 31% of all Testudines species. The concern here is that species with complete trait data are more likely to be well-studied species, since trait values are typically documented through the very research effort the model is trying to explain. This creates a potential circularity: the model may primarily be explaining variation among the already better-known species, and the predictors that emerge as significant may not generalise to the full dataset.

I recognise that this is a common limitation of comparative studies relying on trait data, and that the authors have already taken steps to maximise sample size (e.g. by excluding female age at maturity and generation length). Nevertheless, I would encourage the authors to briefly acknowledge this potential circularity in the Discussion, or alternatively, to provide a simple comparison of mean research effort between species included versus excluded from the multivariate model to quantify the extent of this bias.

Minor points

3. 32 zero-publication species. The increase from 29 to 32 species with no publications (line 322) following the taxonomy update is a striking finding. I would encourage the authors to include the full list of these 32 species as a supplementary table. This would be a valuable resource for the conservation community and would cost very little in terms of additional effort.

4. Country research rank variable. The addition of this variable (lines 185–196) is a welcome response to Reviewer 2’s suggestion, and the results are illuminating: species in the top 10 countries had a mean research effort of 145 articles versus 9 for species absent from the top 20 (lines 354–358). It is interesting, though perhaps not surprising, that this variable was not retained in the multivariate model, likely due to redundancy with the biogeographic realm. The authors note this (response to Reviewer 2), and a brief mention in the Discussion of why these two variables are partially redundant would be useful for the reader.

5. Hypothesis framing. The Introduction continues to frame the research questions as open-ended queries (lines 102–106) rather than as directional predictions with stated rationale. I appreciate that this is an exploratory study and that PLOS ONE’s scope accommodates this approach. Nevertheless, even a brief statement of expected patterns (e.g. “We predicted that species occurring in regions with greater research infrastructure would attract more attention, given...”) would sharpen the framing and strengthen the reader’s ability to evaluate the findings against clear expectations. This is offered as a suggestion rather than a requirement.

Summary

This is a useful and well-executed study that makes a meaningful contribution to our understanding of research biases in Testudines, with clear implications for conservation prioritisation and the interpretation of large-scale comparative analyses. The authors have been responsive to the first round of reviews and have substantially improved the manuscript. The remaining issues are relatively minor. I recommend that the wording error on line 326 be corrected, and that the authors consider briefly addressing the potential bias in the multivariate model’s species subset. The other suggestions above are offered in the spirit of strengthening an already solid piece of work.

.

Reviewer #3: No

Reviewer #4: No

---

## [Author Response · Author response to Decision Letter 2]

25 Mar 2026

Reviewers' comments:

Reviewer #3: The revised manuscript was checked and there is no error found in the text. I checked the comments and the author responses. It is sufficient for publication now.

Reviewer #4: The authors have done commendable work in revising this manuscript. The revisions are substantial and demonstrate a genuine engagement with the concerns raised by both reviewers and the editor. In particular, the updated taxonomy (2025 Turtles of the World checklist, now covering 370 species), the addition of the country research rank variable to address geopolitical bias, the parallel multivariate analyses with and without sea turtles, and the resolution of the clutch size sign reversal through its exclusion from the multivariate model are all welcome improvements. The removal of the problematic review papers figure and the correction of the Triassic/Testudines error also strengthen the manuscript. The paper reads more clearly and is better supported than the first submission.

That said, a small number of issues remain, which I outline below. Most are minor and should be straightforward to address. I believe the manuscript is approaching publication readiness, and I would recommend minor revisions before acceptance.

Our answer: Many thanks for this new thorough review of our article. We are very grateful and have now addressed these minor comments.

Substantiative points

1. Wording error on line 326 (persists from the previous version). The sentence currently reads: "This increase was particularly important in marine turtles, which attracted 18.1 times more research in the 1980s as compared to the 2010s." The direction is reversed. Given the context and Figure 1, the intended meaning is clearly that the 2010s saw 18.1 times more research on marine turtles than the 1980s, not the other way round. This should be corrected to read something like: "...which attracted 18.1 times more research in the 2010s as compared to the 1980s."

Our answer: The reviewer is right, and we changed the text accordingly (lines 326-327).

2. Potential bias in the multivariate model subset. The final multi-predictor model (Table 3) is based on 115 species for which all trait values were available, out of 370 total. This represents roughly 31% of all Testudines species. The concern here is that species with complete trait data are more likely to be well-studied species, since trait values are typically documented through the very research effort the model is trying to explain. This creates a potential circularity: the model may primarily be explaining variation among the already better-known species, and the predictors that emerge as significant may not generalise to the full dataset.

I recognise that this is a common limitation of comparative studies relying on trait data, and that the authors have already taken steps to maximise sample size (e.g. by excluding female age at maturity and generation length). Nevertheless, I would encourage the authors to briefly acknowledge this potential circularity in the Discussion, or alternatively, to provide a simple comparison of mean research effort between species included versus excluded from the multivariate model to quantify the extent of this bias.

Our answer: This is an important point, and we followed the reviewer’s suggestion by both acknowledging this bias in the discussion and including a comparison of the mean research effort for the species included vs excluded from the multivariate model (see lines 491-495).

Minor points

3. 32 zero-publication species. The increase from 29 to 32 species with no publications (line 322) following the taxonomy update is a striking finding. I would encourage the authors to include the full list of these 32 species as a supplementary table. This would be a valuable resource for the conservation community and would cost very little in terms of additional effort.

Our answer: Following the reviewer’s advice, we have now included this list as a supplementary material (Suppl. Mat. 4).

4. Country research rank variable. The addition of this variable (lines 185–196) is a welcome response to Reviewer 2’s suggestion, and the results are illuminating: species in the top 10 countries had a mean research effort of 145 articles versus 9 for species absent from the top 20 (lines 354–358). It is interesting, though perhaps not surprising, that this variable was not retained in the multivariate model, likely due to redundancy with the biogeographic realm. The authors note this (response to Reviewer 2), and a brief mention in the Discussion of why these two variables are partially redundant would be useful for the reader.

Our answer: We added a sentence about this redundancy lines 460-462 of the discussion.

5. Hypothesis framing. The Introduction continues to frame the research questions as open-ended queries (lines 102–106) rather than as directional predictions with stated rationale. I appreciate that this is an exploratory study and that PLOS ONE’s scope accommodates this approach. Nevertheless, even a brief statement of expected patterns (e.g. “We predicted that species occurring in regions with greater research infrastructure would attract more attention, given...”) would sharpen the framing and strengthen the reader’s ability to evaluate the findings against clear expectations. This is offered as a suggestion rather than a requirement.

Our answer: We decided not to include predictions at the end of the introduction as we only had clear expectations for some (e.g., range size, biogeographic realm) but not all of the tested predictors. Indeed, several predictors were considered to test for the existence of potential biases that could affect the conclusions of comparative analyses (e.g., diet, habitat, clutch size…), rather than with clear expectations with regards to differences in research effort.

Summary

This is a useful and well-executed study that makes a meaningful contribution to our understanding of research biases in Testudines, with clear implications for conservation prioritisation and the interpretation of large-scale comparative analyses. The authors have been responsive to the first round of reviews and have substantially improved the manuscript. The remaining issues are relatively minor. I recommend that the wording error on line 326 be corrected, and that the authors consider briefly addressing the potential bias in the multivariate model’s species subset. The other suggestions above are offered in the spirit of strengthening an already solid piece of work.

Our answer: We are very grateful to the reviewer for comments of the two rounds of review that helped us to substantially improve our manuscript.

---

## [Editor Report · Decision Letter 2]

31 Mar 2026

Which turtle should I study? Uneven distribution of research effort across Testudines species

PONE-D-25-36542R2

Dear Dr. Ducatez,

We’re pleased to inform you that your manuscript has been judged scientifically suitable for publication and will be formally accepted for publication once it meets all outstanding technical requirements.

Kind regards,

Patrick R Stephens, Ph.D.

Academic Editor

PLOS One
---

## [Editor Report · Acceptance letter]

PONE-D-25-36542R2

PLOS One

Dear Dr. Ducatez,

I'm pleased to inform you that your manuscript has been deemed suitable for publication in PLOS One. Congratulations! Your manuscript is now being handed over to our production team.

Kind regards,

on behalf of

Dr. Patrick R Stephens

Academic Editor

PLOS One